# Detecting RNA base methylations in single cells by in situ hybridization

Rohan T. Ranasinghe [1], Martin R. Challand [2,4], Kristina A. Ganzinger [1,5], Benjamin W. Lewis [1] Charlotte Softley[1], Wolfgang H. Schmied[3], Mathew H. Horrocks[1], Nadia Shivji[1], Jason W. Chin [1,3], James Spencer[2] & David Klenerman[1]

Methylated bases in tRNA, rRNA and mRNA control a variety of cellular processes, including protein synthesis, antimicrobial resistance and gene expression. Currently, bulk methods that report the average methylation state of $\sim 10^4$–$10^7$ cells are used to detect these modifications, obscuring potentially important biological information. Here, we use in situ hybridization of Molecular Beacons for single-cell detection of three methylations ($m^6_2A$, $m^1G$ and $m^3U$) that destabilize Watson–Crick base pairs. Our method—methylation-sensitive RNA fluorescence in situ hybridization—detects single methylations of rRNA, quantifies antibiotic-resistant bacteria in mixtures of cells and simultaneously detects multiple methylations using multi-color fluorescence imaging.

[1] Department of Chemistry, University of Cambridge, Lensfield Road, Cambridge CB2 1EW, UK. [2] School of Cellular and Molecular Medicine, University of Bristol, Bristol BS8 1TD, UK. [3] Medical Research Council Laboratory of Molecular Biology, Francis Crick Avenue, Cambridge CB2 0QH, UK. [4]Present address: School of Biochemistry, University of Bristol, Bristol BS8 1TD, UK. [5]Present address: Max-Planck-Institut für Biochemie (MPI for Biochemistry), 82152 Martinsried, Germany. Correspondence and requests for materials should be addressed to R.T.R. (email: rr360@cam.ac.uk) or to M.R.C. (email: Martin.Challand@bristol.ac.uk)

t is becoming increasingly clear that methylated RNA bases play key roles in all forms of life. It has been known for over 50 years that ribosomal RNA (rRNA) contains methylated bases[1]: 21 of the 35 post-transcriptional modifications to the *Escherichia coli* ribosome include base methylations[2], while 8 other methylated bases confer antibiotic resistance by preventing binding of protein synthesis inhibitors[3]. For example, the rRNA methylations catalyzed by Erm (forming $N^6$-dimethyladenosine, $m^6_2A$)[4] and Cfr (forming 2,8-dimethyladenosine, $m^{2,8}A$)[5] both render bacteria multi-drug resistant. Among the numerous modifications to transfer RNA (tRNA), 1-methylguanosine ($m^1G$) at position 37, which minimizes frameshift mutations during protein synthesis[6], is notable for being conserved across all kingdoms of life. The observations that the fat mass and obesity-associated (FTO) protein demethylates $N^6$-methyladenosine ($m^6A$), 3-methyluridine ($m^3U$) and 3-methylthymidine ($m^3T$) in single-stranded RNA[7, 8] led to the realization that eukaryotic messenger RNAs (mRNAs) undergo dynamic and widespread methylations at N6 and N1 of adenine[9–13], which likely regulate expression. This flurry of recent discoveries has refocused attention on post-transcriptional RNA modifications[14], and spawned a new field: epitranscriptomics.

The epitranscriptome lay underappreciated until the last 5–10 years because methods for detecting methylated bases were unsuitable for studying mRNA. Conventionally, cellular RNAs are digested to individual nucleosides that are separated by liquid chromatography to identify modified bases, which are then put into sequence context by a separate reverse transcription assay, analyzed using gel electrophoresis[15]. These bulk methods have detection limits in the femtomole to picomole range, which even for abundant species like rRNAs therefore report the average methylation state of $\sim 10^4$–$10^7$ cells[16]. Even state-of-the-art methods for mapping modified bases—based on immunoprecipitation followed by next-generation sequencing—pool the lysates from $> 10^4$ cells[17]. In contrast, microscopy-based fluorescence in situ hybridization (FISH)[18] is revolutionizing transcription profiling by capturing cell-to-cell variation in mRNA levels[19], and revealing the sub-cellular distributions of specific mRNAs on the basis of their unique nucleotide sequences[20]. In situ hybridization techniques capable of detecting methylated bases within specific RNAs could be similarly illuminating, greatly enhancing our understanding of the biological roles of individual methylations.

Here, we use in situ hybridization to detect epitranscriptomic modifications at the single-cell level, exploiting the fact that methyl groups on the Watson–Crick faces can impair base pairing[21, 22]. Our approach deploys hybridization probes that are sensitive to methylation of their complementary RNA sequences as intracellular thermodynamic sensors. We call this method "methylation-sensitive RNA fluorescence in situ hybridization" (MR-FISH), which we validate in a series of methyltransferase-knockout bacterial cell lines, focusing on rRNA methylations. MR-FISH is sensitive to single methylations, and can characterize the composition of heterogeneous mixtures of cells that differ only in RNA methylation.

## Results

### Detecting tetramethylation by KsgA methyltransferase.
In designing a hybridization assay for methylation, we selected Molecular Beacons (Supplementary Fig. 2)[23]. We reasoned that these hybridization probes, which discriminate single-base changes better than linear probes because they can form a hairpin structure rather than bind to a mismatched nucleic acid[24], might also be sensitive to methylation of a complementary nucleic acid. We first tested whether a Molecular Beacon could detect the tetramethylation catalyzed by the methyltransferase (MTase) KsgA (Fig. 1a), one of the best-studied post-transcriptional modifications of rRNA. KsgA dimethylates A1518 and A1519 of 16S rRNA to form consecutive $m^6_2A$ bases, subtly restructuring the 30S subunit of the ribosome;[25] bacteria that lack this modification are resistant to the aminoglycoside antibiotic, kasugamycin[26]. Because the three states of Molecular Beacons (hairpin structure, hybridized to the target and random coil) have different fluorescence intensities due to changes in the fluorophore–quencher distance (Supplementary Fig. 2), the stability of duplexes they form with different target sequences can be measured by monitoring fluorescence during thermal melting. This technique verified that a Molecular Beacon designed to detect methylation by KsgA shows a transition for melting of an RNA/DNA duplex with complementary unmethylated RNA, but only a transition for melting of the hairpin loop in the presence of tetramethylated RNA (Fig. 1c). This indicates that the impaired base pairing of adjacent $m^6_2A$ nucleotides (Fig. 1b) inhibits the binding of the methylation-sensitive probe.

To detect post-transcriptional modifications inside fixed cells, we use two probes: a green-fluorescent (Alexa Fluor 488-labeled) methylation-sensitive probe and a red-fluorescent (Alexa Fluor 647-labeled) probe that binds the same rRNA strand, at a sequence remote from the modification site that does not undergo methylation. This red-labeled methylation-insensitive probe acts as an internal calibrant, meaning that the red/green ratio of each cell—rather than the green intensity alone—indicates the extent of methylation. This ratio distinguishes between cells with a high proportion of methylated rRNA and cells which simply have a low concentration of rRNA (Fig. 1d). To investigate whether MR-FISH distinguishes between cells with highly methylated and unmethylated ribosomes, we used an *E. coli* which expresses all constitutive rRNA MTases (parent strain BW25113), and a mutant of this strain in which the KsgA MTase is deleted ($\Delta ksgA$), confirming the methylation state of each cell type by high-performance liquid chromatography (HPLC; Supplementary Fig. 3). Both bacteria fluoresce red due to the methylation-insensitive probe, but only the $\Delta ksgA$ cells, with A1518 and A1519 of 16S rRNA unmethylated, are brightly stained green by the methylation-sensitive probe (Fig. 1e). Automated analysis of the wide-field epifluorescence images (Methods, Supplementary Fig. 4) allows us to extract fluorescence intensities of hundreds of single cells in both color channels, which quantitatively confirms that MR-FISH distinguishes between cells with methylated and unmethylated ribosomes (Fig. 1f). Timecourses of hybridization show that $\Delta ksgA$ cells can be distinguished from the parent strain after 30 min, but that they are best discriminated after $\geq 4.5$ h, when the ratio of their green fluorescence intensities is $> 10$ (Supplementary Fig. 5). The fluorescence of the methylation-sensitive probe is selectively reduced in the presence of excess unlabeled oligonucleotide with the same sequence, confirming that the signal results primarily from hybridization, rather than non-specific binding to cellular components or autofluorescence (Supplementary Fig. 10). When the red/green ratios of individual cells are extracted from MR-FISH image data, each cell type displays a log-normal distribution (Fig. 1g and Supplementary Fig. 6), which unambiguously discriminates between cells with tetramethylated and unmethylated ribosomes in replicate measurements (Fig. 1h and Supplementary Fig. 7).

### MR-FISH is sensitive to single methylations.
We then challenged MR-FISH to detect single methylations of rRNA using the same parent strain and two additional mutants, from which two other constitutive MTases are deleted: RrmA and RsmE, which respectively catalyze introduction of 1-methylguanine ($m^1G$) at G745 in 23S rRNA[27], and 3-methyluridine ($m^3U$) at U1498 in

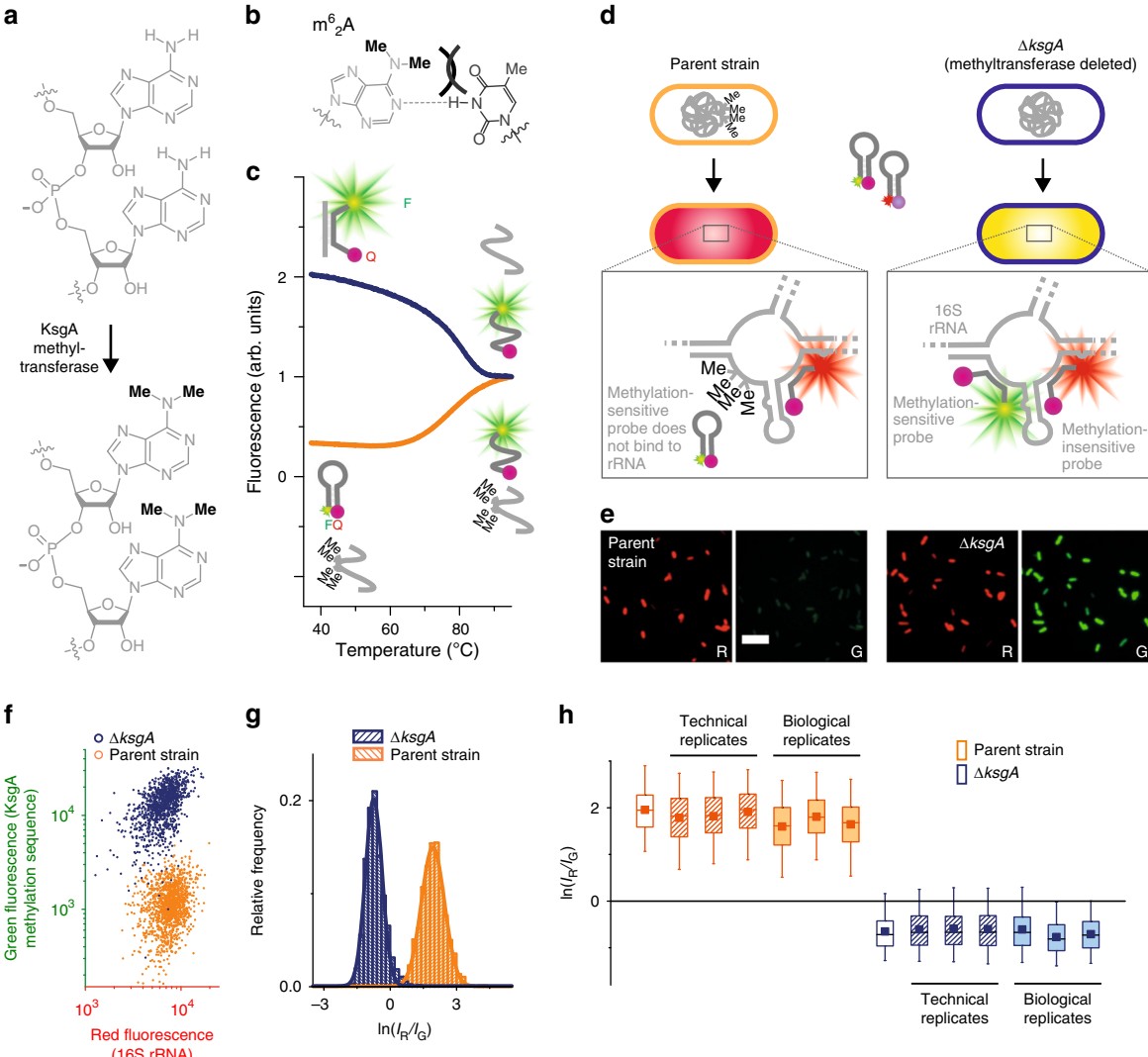

**Fig. 1** Destabilized base pairing by methylated RNA enables detection of post-transcriptional modification in single cells. **a** $N^6$-Dimethylation of A1518 and A1519 in 16S rRNA catalyzed by KsgA. **b** $N^6$-Dimethyladenine ($m^6_2A$) is sterically inhibited from base pairing with thymine. **c** A Molecular Beacon can discriminate between methylated and unmethylated synthetic RNA, as revealed by in vitro thermal melting. **d** MR-FISH assay for rRNA methylation, using two Molecular Beacons and two strains of E. coli:, one expressing active KsgA (parent strain) and one mutant strain with the methyltransferase deleted from its genome (ΔksgA). **e** Fluorescence images showing that ΔksgA E. coli are stained red and green by MR-FISH probes, while parent strain cells are brightly stained only by the methylation-insensitive red probe. Scale bar: 10 μm. **f** Scatter plots of red and green fluorescence intensities of parent strain (n = 1315 cells) and ΔksgA (n = 1112 cells) bacteria stained by MR-FISH. **g** Log-normally distributed ratios of red fluorescence intensity ($I_R$) to green fluorescence intensity ($I_G$) of bacteria stained by MR-FISH. **h** Box plots (box: 25th–75th percentiles; whiskers: 5th–95th percentiles; horizontal line: medians; squares: means) of two-color ratios shown in (**g**) (empty bars), as well as technical replicates (hatched bars) and biological replicates (filled bars). From left to right, n = 1315, 1056, 1173, 1115, 865, 1253, 927, 1110, 1291, 1523, 1374, 1118, 850 and 988 cells per sample

16S rRNA[28]. Methylation-sensitive probes for each modification discriminate well between methylated and unmethylated synthetic RNAs, but unlike the tetramethylated KsgA target sequence, fluorescence melting showed evidence of duplex formation between both methylation-sensitive beacons and their singly methylated RrmA and RsmE target sequences (Fig. 2a, b). The duplex formed by the RsmE target containing a single $m^3U$ base was stable enough to thermodynamically characterize, showing it to be destabilized by ~20 kJ mol$^{-1}$ compared with its unmethylated counterpart, which decreases its melting temperature by ~17 °C (Supplementary Fig. 2). Nevertheless, MR-FISH detected both methylated bases inside fixed cells, although a higher formamide concentration (30% vs. 20%) was required to discriminate the less-destabilizing $m^3U$ modification (Fig. 2c–f, Supplementary Figs 8 and 9). These data show that MR-FISH can detect even single methylations of RNA.

**Characterizing mixtures of cells with different methylation states.** A major advantage of detecting epitranscriptomic modifications on a cell-by-cell basis, rather than in lysates pooled from thousands or millions of cells, is the potential to study heterogeneous populations without separating them first. To demonstrate this concept, we used MR-FISH to characterize the composition of mixtures of E. coli and the kasugamycin-resistant derivative ΔksgA. We first combined different ratios of separately cultured samples of each strain and analyzed the mixtures using MR-FISH. Fitting the resulting two-color-ratio histograms with bimodal Gaussian distributions returns compositions that are in excellent agreement with the known proportions of parent strain and ΔksgA cells (Fig. 3a–c). Next, we measured the composition of a co-culture of these bacteria using MR-FISH, which was in good agreement with that determined by colony counting (Fig. 3d–g). The greater precision of our single-cell approach—

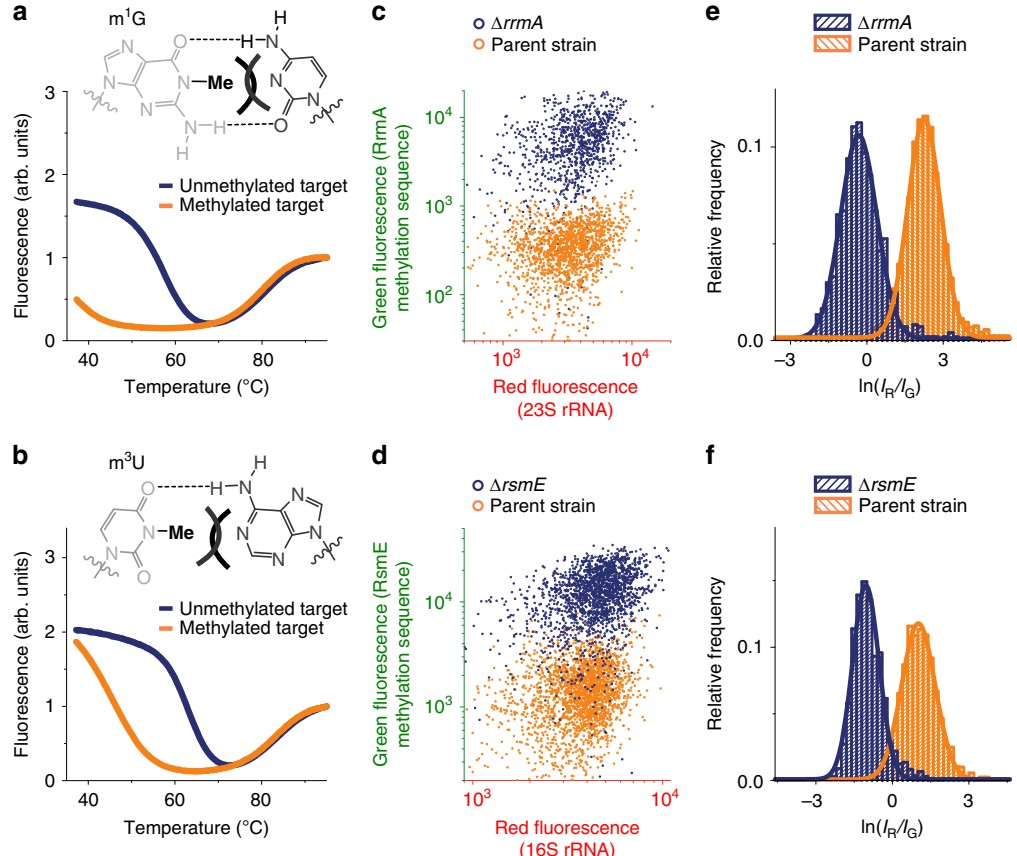

**Fig. 2** MR-FISH can detect single methylated m$^1$G and m$^3$U bases. Discrimination by Molecular Beacons between methylated and unmethylated synthetic RNA target sequences of **a** RrmA and **b** RsmE. Scatter plots of red and green fluorescence intensities of bacteria stained by MR-FISH using hybridization probes for RrmA (**c**; $n = 1567$ (parent strain) and 966 ($\Delta rrmA$) cells) and RsmE (**d**; $n = 2266$ (parent strain) and 1894 ($\Delta rsmE$) cells) modifications. Log-normally distributed ratios of red fluorescence intensity ($I_R$) to green fluorescence intensity ($I_G$) of bacteria stained by MR-FISH using hybridization probes for **e** RrmA and **f** RsmE

reflected in the smaller standard deviation of the measured fraction (Fig. 3g)—is in line with that expected from Poisson statistics, because 450–850 cells are counted per measurement compared with 25–100 colonies.

Finally, we combined methylation-sensitive Molecular Beacons to demonstrate simultaneous detection of two methylations using multicolor imaging (Fig. 4). We multiplexed MR-FISH using a probe cocktail comprising an Alexa-Fluor-647-labeled-methylation-insensitive probe, an Alexa-Fluor-488-labeled probe sensitive to KsgA methylation and a Cy3-labeled probe sensitive to RrmA methylation (Fig. 4a). A mixture of bacteria containing either of these methylations ($\Delta ksgA$ or $\Delta rrmA$) or both methylations (parent strain) are therefore stained with different color combinations: either Alexa Fluor 647 and Alexa Fluor 488 ($\Delta ksgA$), Alexa Fluor 647 and Cy3 ($\Delta rrmA$), or Alexa Fluor 647 only (parent strain) (Fig. 4b). Comparison to pure samples of each bacterium confirms that MR-FISH can accurately identify the components of a complex mixture of bacteria with multiple different methylation states (Fig. 4c, d and Supplementary Fig. 11).

## Discussion

This work demonstrates the detection of methylated bases in RNA at the single-cell level. MR-FISH is sensitive to single methylations, and can characterize the composition of mixtures of cells with different RNA methylation states. Our results show excellent discrimination between cells with and without specific methylations; further studies will establish the sensitivity of MR-FISH to different degrees of methylation. Three different methylations have been detected here, but in its current form, our hybridization-based approach will not be applicable to modifications that do not affect Watson–Crick base pairing, such as 5-methylcytidine (m$^5$C) or pseudouridine. On the other hand, m$^1$A has recently been shown to substantially disrupt RNA duplexes[29], while m$^6$A forms a "spring-loaded" base pair containing a steric clash between the N6-methyl group and N7[22]. Encouragingly, we have confirmed that the presence of m$^6$A does destabilize duplex formation with a Molecular Beacon, but to a lesser degree than the other modifications detected here (Supplementary Fig. 12).

The fact that standard epifluorescence microscopes can be used to detect methylation of abundant RNAs (Supplementary Fig. 13) as well as its simplicity (Supplementary Fig. 14) makes MR-FISH accessible to wide range of users. For example, our method could enable basic research into antibiotic resistance conferred by methylated bases in single or mixed bacterial populations. In particular, MR-FISH could reveal the extent of methylation needed for resistance and stochastic cell-to-cell variation within resistant populations, as well as offering a way to screen methyltransferase inhibitors inside cells with a molecular readout. The fact that MR-FISH requires simple optical equipment, is capable of multiplexing and can identify specific rRNA methylations in cells after only 30 min also highlights its potential in diagnostic tests for antibiotic-resistant bacteria. In future applications, the abundance of the RNA of interest will determine the most appropriate imaging mode. In exponentially growing

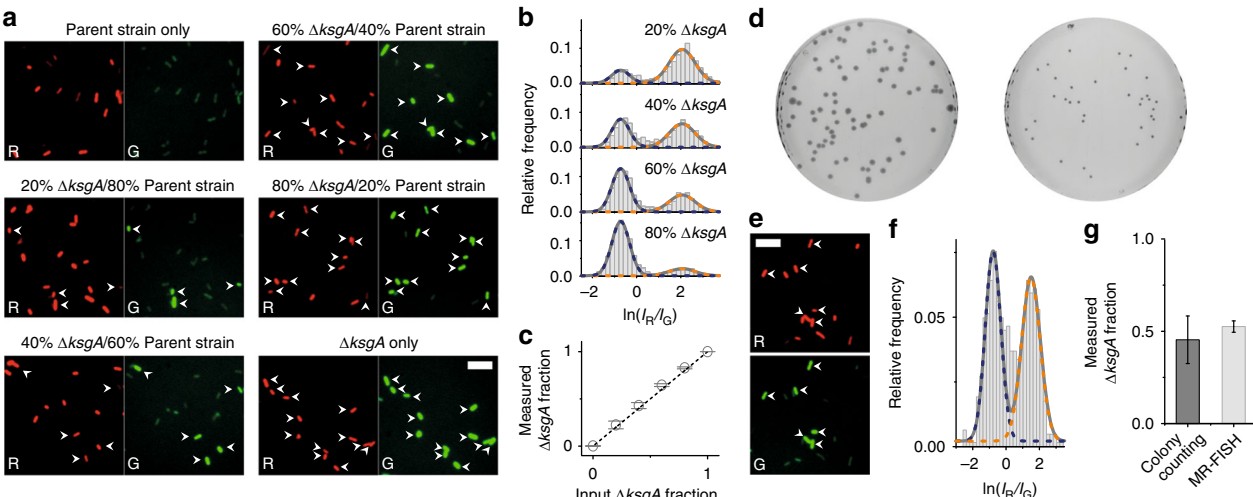

**Fig. 3** Detecting heterogeneity in mixtures of bacteria with different rRNA methylation status. **a** False-colored red and green fluorescence images of pure samples and mixtures of parent strain and Δ*ksgA E. coli* after MR-FISH. All red images are shown at one fixed brightness and contrast, and all green images at another fixed brightness and contrast. To guide the eye, manually selected bacteria with high green fluorescence intensity are indicated with arrows in both red and green images. Scale bar: 10 µm. **b** Histograms of two-color ratios obtained from mixtures of parent strain and Δ*ksgA* bacteria (n = 1372, 1639, 1593 and 1183 cells respectively for 20%, 40%, 60% and 80% Δ*ksgA*). The data are fitted to a bimodal Gaussian distribution with centers and widths defined by fits of MR-FISH data from pure samples of each bacterium. **c** Quantification of the fraction of Δ*ksgA E. coli* in mixtures with parent strain bacteria, based on the areas of each population extracted from the bimodal distributions of two-color ratios. The means and standard deviations of three technical replicates are plotted, while the equation of the dotted line is "measured fraction=input fraction". **d** Growth of co-culture of parent strain and Δ*ksgA E. coli* on an agar plate (left) and a plate containing kanamycin (right), which selects for Δ*ksgA E. coli* only. **e** False-colored red and green fluorescence images of co-culture after MR-FISH. Scale bar: 10 µm. **f** Histogram of two-color ratios of the co-culture (n = 625 cells), fitted to a bimodal Gaussian distribution with centers and widths defined as in (**b**). **g** Comparison of the fraction of Δ*ksgA E. coli* in co-culture as measured by colony counting and MR-FISH. The means and standard deviations from each technique are plotted (n = 5 technical replicates)

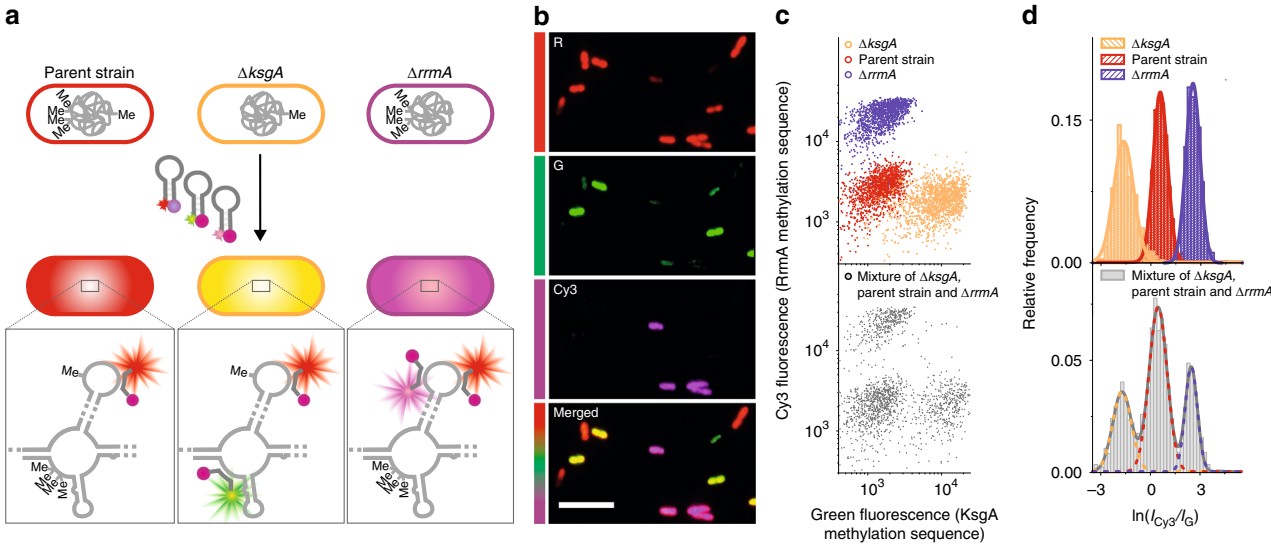

**Fig. 4** Multiplexed detection of rRNA methylations with MR-FISH. **a** Multiplexed assay for two rRNA methylations, using three Molecular Beacons, staining an unmethylated portion of the 23S rRNA (Alexa Fluor 647-labeled), the KsgA methylation sequence (Alexa Fluor 488-labeled) and the RrmA methylation sequence (Cy3-labeled). Three strains of *E. coli* are used: one expressing active KsgA and RrmA (parent strain), one mutant strain with the KsgA methyltransferase deleted from its genome (Δ*ksgA*) and another mutant strain with the RrmA methyltransferase deleted from its genome (Δ*rrmA*). **b** False-colored fluorescence images of a mixture of parent strain, Δ*ksgA* and Δ*rrmA E. coli* stained with the cocktail of Molecular Beacons. Scale bar: 10 µm. **c** Scatter plots of green and Cy3 fluorescence intensities of parent strain (n = 1362 cells), Δ*ksgA* (n = 1411 cells) and Δ*rrmA* (n = 1288 cells) bacteria assayed separately and in a mixture (n = 1659 cells). **d** Log-normally distributed ratios of Cy3 fluorescence intensity ($I_{Cy3}$) to green fluorescence intensity ($I_G$) of bacteria stained by MR-FISH in (**c**)

bacteria as measured here, rRNAs are present at ~50,000 copies/cell (~50 µM)[30], which enables single-cell measurements with epifluorescence microscopy. Individual bacterial tRNAs are present at concentrations of 1−30 µM[31], so should be detectable in

the same way. At the other end of the scale, detecting any specific endogenous mRNA—typically present at <1−30 copies per eukaryotic cell[32, 33]—inherently requires single-molecule implementation of MR-FISH, which could be used to investigate

**Table 1 Sequences of synthetic oligonucleotides**

| Code name | Sequence (5′–3′) |
|---|---|
| RNAs | |
| KsgA UM | GUAA*CCGUAGGGG**AA**CCUGCGG*UUGGAUCAC |
| KsgA Me | GUAA*CCGUAGGGG**m⁶₂Am⁶₂**ACCUGCGG*UUGGAUCAC |
| RrmA UM | GAA*CCGACUAAU**G**UUGAAAAA*UUAGCG |
| RrmA Me | GAA*CCGACUAAU**m¹G**UUGAAAAAUUAGCG |
| RsmE UM | GGGUG*AAGUCG**U**AACAAGG*UAACCG |
| RsmE Me | GGGUG*AAGUCG**m³U**AACAAGG*UAACCG |
| RlmJ UM | UUGAACU*CGCUGUG**A**AGAUGC*AGUGUACCCG |
| RlmJ Me | UUGAACU*CGCUGUG**m⁶A**AGAUGC*AGUGUACCCG |
| DNAs | |
| KsgA MB | Alexa Fluor 488-<u>CCGCC</u>*CCGCAGG**TT**CCCCTACGG*<u>CGG</u>-DABCYL |
| RrmA MB | Alexa Fluor 488-<u>CCGCCG</u>*TTTTTCAA**C**ATTAGT*<u>CGGCGG</u>-DABCYL |
| RrmA MB-Cy3 | Cy3-<u>CCGCCG</u>*TTTTTCAA**C**ATTAGT*<u>CGGCGG</u>-BHQ2 |
| RsmE MB | Alexa Fluor 488-<u>CGCGCC</u>*TTGTT**A**CGACTTC*<u>GGCGCG</u>-DABCYL |
| RlmJ MB | FAM-<u>CGCGC</u>*ATCT**T**CACAGC*<u>GCG</u>-DABCYL |
| 16S MB | Alexa Fluor 647-<u>CACTCC</u>*GCTGCCTCCCGTAG*<u>GAGTG</u>-BHQ1 |
| 16S MB-G | Alexa Fluor 488-<u>CACTCC</u>*GCTGCCTCCCGTAG*<u>GAGTG</u>-DABCYL |
| 23S MB | Alexa Fluor 647-<u>CGGCG</u>*AGCAAGTCGCTTCACCTACATAT*<u>CGCCG</u>-BHQ1 |
| KsgA BLK | <u>CCGCC</u>*CCGCAGG**TT**CCCCTACGG*<u>CGG</u> |
| RrmA BLK | <u>CCGCCG</u>*TTTTTCAA**C**ATTAGT*<u>CGGCGG</u> |
| RsmE BLK | <u>CGCGCC</u>*TTGTT**A**CGACTTC*<u>GGCGCG</u> |
| HLP1 | AGTGGTAAGCGCCCTCCCGA |
| HLP2 | ACCCCAGTCATGAATCACAA |
| HLP3 | GCAGGTTCCCCTACGGTTA |
| HLP4 | TAAGGAGGTGATCCAACC |

RNA nucleotides complementary to MB sequences are shown in italics, with methylated bases or bases whose methylation is catalyzed by MTase shown in bold italics. DNA nucleotides complementary to rRNA sequences are shown in italics, with nucleotides complementary to bases methylated by the relevant methyltransferase in bold italics. The self-complementary sections of Molecular Beacons that fold into the stem are underlined. The structures of modified nucleosides, fluorophores and quenchers are given in Supplementary Fig. 1

epitranscriptomic variation between cells and how methylation influences the spatial distribution of mRNA.

MR-FISH has distinct advantages and disadvantages compared with existing techniques. While the need for pre-existing knowledge of the locations of methylated bases to design methylation-sensitive probes and the finite multiplexing capacity of FISH make it unsuitable for transcriptome-wide discovery of base methylations, MR-FISH offers the ability to analyze methylation in large numbers of single cells, including heterogeneous populations and/or tissue samples. Alongside sequencing-based techniques that identify modifications[17], and chemical probes that establish their structural effects[34], MR-FISH promises to add another dimension to studies of methylated bases, by capturing cell-to-cell variation and potentially revealing the sub-cellular locations of specific methylated RNAs.

## Methods

**Materials**. The *E. coli* strain BW25113 (parent strain) was purchased from ThermoFisher Scientific (Ulm, Germany). Three derivatives of this strain: JW0050-3 with specific deletion of the *ksgA* gene (Δ*ksgA*); JW1181-1 with specific deletion of the *rrmA* gene (Δ*rrmA*); and JW2913-1 with specific deletion of the *rsmE* gene (Δ*rsmE*), were sourced from the Coli Genetic Stock Centre (Yale, USA)[35, 36]. Bacterial growth medium was purchased from Appleton Woods (Birmingham, UK) and was prepared by dissolving in deionized water followed by autoclaving at 121 °C for 20 min. Phosphate-buffered saline (PBS) tablets, sodium chloride, tris (hydroxymethyl)aminomethane hydrochloride (Tris-HCl), sodium dodecyl sulfate (SDS), diethyl pyrocarbonate (DEPC), formamide, benzamidine, phenylmethylsulfonyl fluoride (PMSF), magnesium chloride, ammonium chloride, kanamycin, paraformaldehyde, 2-mercaptoethanol and sucrose were purchased from Sigma Aldrich (Gillingham, UK). The 125:24:1 phenol/chloroform/isoamyl alcohol mixture and chloroform were purchased from ThermoFisher Scientific (Paisley, UK). P1 nuclease, bacterial alkaline phosphatase, unmethylated nucleoside standards (A, C, G and U) were purchased from Sigma Aldrich, methylated nucleoside standards ($N^6$-dimethyladenosine, 1-methylguanosine and 3-methyluridine) from Carbosynth (Compton, UK) and HPLC-grade solvents from VWR International Ltd (Lutterworth, UK). To avoid degradation of RNA, stock solutions of all buffer components (except formamide) for fluorescence melting and FISH were prepared using 0.1% DEPC to inactivate RNases and then autoclaved to decompose DEPC prior to use. All microcentrifuge tubes and pipette tips used were

certified RNase free and bench surfaces cleaned with RNaseZAP™ according to the manufacturer's instructions (Sigma Aldrich). The sequences and code names of all oligonucleotides used in this work are given in Table 1, while structures of modified nucleosides, fluorophores and quenchers are given in Supplementary Fig. 1. RNA oligonucleotides (synthesized on the 1 μmol scale and purified by denaturing polyacrylamide gel electrophoresis) were purchased from Dharmacon/GE Healthcare (Little Chalfont, UK); their identities were also confirmed by digestion to nucleosides followed by HPLC (Supplementary Fig. 3). Molecular Beacons (synthesized on the 0.2 μmol scale and purified by double HPLC) were designed with the aid of the mfold and DINAMelt web servers[37–39], and purchased from ATDBio (Southampton, UK). Unlabeled DNA oligonucleotides (synthesized on either 0.2 or 1 μmol scale and purified by HPLC) were purchased from Sigma Aldrich. Lyophilized oligonucleotides received from the suppliers were dissolved in DEPC-treated deionized water to concentrations of 20–100 μM as confirmed by $OD_{260}$ (Nanodrop 2000, ThermoFisher Scientific) and aliquots stored at −20 °C.

**Thermodynamic characterization of methylation-sensitive Molecular Beacons**. Fluorescence melting curves of Alexa-Fluor-488-labeled-methylation-sensitive probes (concentrations: 2.5 μM (KsgA MB and RlmJ MB) or 2 μM (RrmA MB and RsmE MB)) in the presence and absence of methylated and unmethylated RNA targets (concentrations: 0, 5, 10, 15, 20, 30 and 35 μM (KsgA UM and KsgA Me), or 0, 25, 30, 35, 40, 45 and 50 μM (RrmA UM, RrmA Me, RsmE UM and RsmE Me), or 0, 15, 20, 25, 30, 35 and 40 μM (RlmJ UM and RlmJ Me) in buffer (1× PBS (KsgA MB) or 1× PBS with 1 M NaCl (RrmA MB, RsmE MB and RlmJ MB)) were recorded using a Roche LightCycler® 480, using the fluorescein filter set (exciting at 465 nm and measuring fluorescence at 510 nm). Samples were prepared in a 96-well plate (LightCycler® 480 Multiwell Plate 96, white, Roche, West Sussex, UK) in a total volume of 20 μL. For experiments in the absence of RNA targets (i.e., Molecular Beacon only), samples were heated to 95 °C, held at this temperature for 10 min, cooled from 95 °C to 37 °C at a rate of 1 °C min⁻¹, held at 37 °C for 10 min and heated from 37 °C to 95 °C at a rate of 1 °C min⁻¹. For experiments in the presence of RNA targets, slower rates of heating/cooling were required to avoid hysteresis: samples were heated to 95 °C, held at this temperature for 10 min and then cooled in steps of 0.2 °C at 0.5 °C min⁻¹ to 37 °C. Samples were then held at 37 °C for 10 min before being heated to 95 °C, again in steps of 0.2 °C at a rate of 0.5 °C min⁻¹. The fluorescence intensity was recorded for both the cooling (annealing) and heating (melting) phases and compared to confirm minimal hysteresis. Five technical replicates were performed under each condition.

Melting temperatures ($T_b$ or $T_d$) were extracted using a script written in MATLAB. For each melting curve, an approximate first derivative was calculated, to which a single-term Gaussian function was fitted, with start and end points

determined by user input and fit quality assessed by visual inspection. The center of the fitted Gaussian was taken to be the melting temperature.

The thermodynamics of hairpin melting (transition $2 \rightarrow 3$, Supplementary Fig. 2a) were analyzed as in Bonnet and Tyagi[24], using the variation of fluorescence intensity with temperature (in the absence of target) to find the equilibrium constant at each temperature, using Eq. 1 (where $F$ is the fluorescence measured at a given temperature, $\alpha$ is the fluorescence of the hairpin structure (the fluorescence measured at 37 °C) and $\beta$ is the fluorescence of the random coil structure (fluorescence measured at 95 °C), Supplementary Fig. 2d).

$$K_{2-3} = \frac{F - \alpha}{\beta - F},$$ (1)

$$R \ln\left(\frac{F - \alpha}{\beta - F}\right) = R \ln K_{2-3} = \Delta H^\circ_{2-3} \frac{1}{T} + \Delta S^\circ_{2-3}.$$ (2)

The equilibrium constant in this form can be used to find the thermodynamic parameters for the transition, assuming no variation of thermodynamic parameters with temperature and no populated intermediates between the hairpin and random coil structures. A plot in the form of Eq. 2 (where $T$ is the temperature in K) was used to calculate the standard enthalpies ($\Delta H^\circ_{2-3}$ in J mol$^{-1}$) and entropies ($\Delta S^\circ_{2-3}$ in J mol$^{-1}$ K$^{-1}$) of the hairpin melting transitions by linear fitting in Excel (Microsoft Corporation, Redmond, Washington, USA) or OriginPro (OriginLab Corporation, Northampton, Massachusetts, USA), and hence $\Delta G^\circ_{2-3}$ (Supplementary Fig. 2e and g).

Thermodynamic parameters for the duplex to hairpin transition (transition $1 \rightarrow 2$, Supplementary Fig. 2a) were calculated as in Bonnet and Tyagi[24]. A plot in the form of Eq. 3 (where $R$ is the gas constant, $T_0$ is the target concentration, $B_0$ is the beacon concentration, $T_d$ is the melting temperature for this transition) was used to calculate the standard enthalpies ($\Delta H^\circ_{1-2}$ in J mol$^{-1}$) and entropies ($\Delta S^\circ_{1-2}$ in J mol$^{-1}$ K$^{-1}$) of the hairpin melting transitions by linear fitting in Excel or OriginPro (Supplementary Fig. 2c and f).

$$R \ln(T_0 - 0.5B_0) = -\Delta H^\circ_{1-2} \frac{1}{T_d} + \Delta S^\circ_{1-2}.$$ (3)

**Bacterial culture and fixation.** Biological replicates were prepared by streaking a glycerol stock onto LB agar and individual colonies picked and cultured overnight at 37 °C in LB broth, supplemented with kanamycin (30 µg mL$^{-1}$) for growth JW0050-3 ($\Delta ksgA$), JW1181-1 ($\Delta rrmA$) and JW2913-1 ($\Delta rsmE$). The overnight culture was subsequently used as a 1% inoculum into fresh LB broth without antibiotics which was incubated at 37 °C in an orbital shaker until the OD$_{600}$ was 0.5–0.7. One volume of 4% formaldehyde (freshly prepared from paraformaldehyde) in PBS (pH 6.9) was then added directly to bacterial culture (7.5 mL, approximately $4.5 \times 10^9$ colony-forming units) and the resulting suspension incubated at room temperature for 90 min. Inactivated bacterial cells were then pelleted by centrifugation, washed three times in PBS (10 mL) and suspended in 50% ethanol/PBS (1 mL) yielding approximately $4.5 \times 10^9$ fixed cells per mL, which were stored at −20 °C until required.

**Bacterial co-culture and colony counting.** Single colonies of BW25113 (parent strain) and JW0050-3 ($\Delta ksgA$) were picked and cultured separately overnight at 37 °C in LB broth, without antibiotics. The overnight cultures were used to inoculate a single flask containing 100 mL of sterile LB broth without antibiotics to a starting OD$_{600}$ of 0.005 absorbance units for each strain. The mixed culture was then incubated at 37 °C in an orbital shaker until the OD$_{600}$ was 0.5–0.7, at which point 7.5 mL of culture was removed and fixed as described above. The ratio of parent/$\Delta ksgA$ was determined by diluting the culture by a factor of $10^6$ in sterile LB broth and spreading 200 µL onto LB agar plates with or without 30 µg mL$^{-1}$ kanamycin. Colonies were counted after 18 h of incubation at 37 °C.

**16S ribosomal RNA purification.** Single colonies BW25113 (parent strain) and JW0050-3 ($\Delta ksgA$) were picked and grown overnight at 37 °C in LB broth, supplemented with kanamycin (50 µg mL$^{-1}$) for growth of the $\Delta ksgA$ strain. The overnight culture was used as a 1% inoculum into pre-warmed LB medium (4 × 1000 mL in 2 L flasks) and cultured without antibiotics until the OD$_{600}$ was 0.5–0.6. The cells were harvested and pellets washed twice by resuspension in PBS (50 mL), prior to being flash-frozen for storage at −80 °C. The ribosome purification protocol was modified from a previously published method[40]. All buffers were supplemented with 0.1 mM benzamidine and 1 mM PMSF directly before use. Cell pellets were thawed on ice, resuspended in 15 mL buffer C (20 mM Tris pH 7.5, 10 mM MgCl$_2$, 200 mM NH$_4$Cl, 6 mM 2-mercaptoethanol) and lysed using an Emulsiflex-C3 homogenizer (Avestin, CAN). The resulting lysate was cleared by pelleting debris twice for 30 min at 30,000×$g$ at 4 °C in an F21-8×50y rotor (Thermo Fisher). The resulting supernatant was layered on a 40 mL sucrose cushion (buffer C adjusted to 500 mM NH$_4$Cl, 1.1 M sucrose) and ribosomes pelleted for 18 h at 142,000×$g$ at 4 °C in a 45 TI rotor (Beckman Coulter). Pellets were washed twice and resuspended for 1 h under gentle agitation in 1 mL buffer D (buffer C adjusted to 1 M NH$_4$Cl). In order to separate the 70S ribosomes into 30S and 50S subunits, the samples were dialyzed overnight at 4 °C against 20 mM Tris

pH 7.5, 1 mM MgCl$_2$, 200 mM NH$_4$Cl, 6 mM 2-mercaptoethanol. Six 10–30% sucrose gradients in buffer D were prepared for each sample, using a Gradient Station IP (Biocomp, USA). Each gradient was prepared in advance, layered with 100–150 µL of sample (~150 A$_{260}$ units) and spun at 58,200×$g$ in a SW28 rotor (Beckman Coulter) for 18 h at 4 °C. Fractions corresponding to the 30S and 50S subunits were separately diluted to 60 mL in buffer C and pelleted for 18 h at 142,000×$g$ at 4 °C in a TI 45 rotor. The pellets were resuspended in 250 µL RNase-free water and flash-frozen for storage. Purified 30S ribosomal subunits were washed twice with 1 vol. of 125:24:1 phenol/chloroform/isoamyl alcohol pH 4.5 followed by once with 1 vol. chloroform. RNA was ethanol precipitated by adding 0.1 vol. of 3 M sodium acetate followed by 2 vol. of ice-cold ethanol and pelleted by centrifugation in a bench top microcentrifuge (10 min, maximum speed). The RNA was dissolved in water (40 µL) and used immediately in RNase digestion reactions.

**RNase digestion.** Purified rRNA (9–50 ng µL$^{-1}$) or synthetic oligonucleotides (25 µM) were dissolved in water. Digestion reactions were prepared by addition of ammonium acetate buffer (20 mM, pH 7) and P1 nuclease (0.5 U), followed by incubation at 45 °C for 4 h. The reaction was then cooled to room temperature, bacterial alkaline phosphatase (150 U) added followed by incubation at 37 °C for a further 2 h. Proteins were precipitated by addition of perchloric acid (1% final vol.) and pelleted by centrifugation (20 min, maximum speed). Supernatants were immediately analyzed by HPLC.

**HPLC analysis.** A fully automated Shimadzu Prominence platform, operated through LabSolutions software, was used for all HPLC analyses. The platform consisted of a LC-20AD ternary, low-pressure mixing solvent delivery system with a DGU-20A inline degasser, SIL-20AHT variable volume auto-injector and auto-sampler, SPD-M20A diode array detector and CTD-10ASVP heated column oven. HPLC-grade solvents were used for preparing the mobile phase. Peaks were identified by comparing their retention times with standards prepared from commercially sourced nucleosides. For analysis of digested and undigested synthetic RNA oligonucleotides, the HPLC conditions shown in Table 2 were used. For nucleoside quantification in digested samples, the integrated peak area was compared to a dilution series prepared from commercially sourced nucleosides. For analysis of digested 16S ribosomal RNA, the HPLC conditions shown in Table 3 were used.

**Fluorescence in situ hybridization.** The protocol for FISH was adapted from Valm et al.[41]. Fixed *E. coli* cells were first washed with wash buffer A (0.9 M NaCl, 0.02 M Tris, pH 7.5, 0.01% SDS and 20% v/v formamide) and resuspended at $6.75 \times 10^6$ cells per µL in 20 µL hybridization buffer (0.9 M NaCl, 0.02 M Tris, pH 7.5, 0.01% SDS, formamide (20% v/v for detecting KsgA and RrmA modifications, 30% v/v for detecting RsmE modification), Alexa-Fluor-647-labeled-methylation-insensitive Molecular Beacon (1 µM 16S MB for detecting KsgA and RsmE modifications, 1 µM 23S MB for detecting RrmA modification), Alexa-Fluor-488-labeled-methylation-sensitive Molecular Beacon (1 µM KsgA MB, 1 µM RrmA MB or 2 µM RsmE MB); unlabeled helper oligonucleotides that bind to rRNA, adjacent to the methylation-sensitive probe, were included for detection of RsmE modification (2 µM HLP1, HLP2, HLP3 and HLP4); for experiments to test whether green fluorescence results from hybridization (Supplementary Fig. 10), unlabeled

**Table 2 HPLC conditions for analysis of digested and undigested synthetic RNA oligonucleotides**

Column: Waters XSelect HSS T3, 100 Å, 3.5 µm, 3.0 × 150 mm

Column temperature: 30 °C

Flow rate: 0.7 mL min$^{-1}$

Injection volume: 8 µL

| Step time / min | Mobile phase composition / % | | |
| --- | --- | --- | --- |
| | A (water) | B (acetonitrile) | C (0.2 M NH$_4$OAc, pH 6, 25% acetonitrile) |
| 0 | 89 | 1 | 10 |
| 2.5 | 89 | 1 | 10 |
| 5.5 | 80 | 10 | 10 |
| 10 | 40 | 50 | 10 |
| 10.1 | 10 | 80 | 10 |
| 11 | 10 | 80 | 10 |
| 11.1 | 89 | 1 | 10 |
| 20 | 89 | 1 | 10 |

### Table 3 HPLC conditions for analysis of digested ribosomal RNA

**Column: Waters XSelect HSS T3, 100 Å, 3.5 µm, 3.0 × 150 mm**

**Column temperature: 30 °C**

**Flow rate: 0.7 mL min$^{-1}$**

**Injection volume: 8 µL**

| Step time / min | Mobile phase composition / % | | |
|---|---|---|---|
| | A (water) | B (acetonitrile) | C(0.2 M NH$_4$OAc, pH 6, 2.5% acetonitrile) |
| 0 | 90 | 0 | 10 |
| 3.6 | 90 | 0 | 10 |
| 6.0 | 88 | 2 | 10 |
| 7.5 | 85 | 5 | 10 |
| 13.5 | 70 | 20 | 10 |
| 24 | 55 | 35 | 10 |
| 27.5 | 10 | 80 | 10 |
| 28.8 | 10 | 80 | 10 |
| 29.0 | 90 | 0 | 10 |
| 38.0 | 90 | 0 | 10 |

Molecular Beacons were added (10 µM KsgA BLK, 10 µM RrmA BLK or 20 µM RsmE BLK)). For three-color multiplexed detection of KsgA and RrmA methylations, the hybridization buffer contained Alexa-Fluor-647-labeled-methylation-sensitive Molecular Beacon (1 µM 23S MB) and two methylation-sensitive Molecular Beacons (1 µM Alexa-Fluor-488-labeled KsgA MB and 1 µM RrmA MB-Cy3). To assess the influence of hybridization stoichiometry on cell-to-cell variation of the two-color ratio (Supplementary Fig. 6), hybridization buffer containing two methylation-insensitive Molecular Beacons was used (1 µM Alexa-Fluor-647-labeled 16S MB and 1 µM Alexa-Fluor-488-labeled 16S MB-G). Samples were incubated at 37 °C for the appropriate length of time (0.5, 1, 2, 3, 4.5, 6, 11, 16 or 21 h for the timecourse data shown in Supplementary Fig. 5, 21 h for all other experiments detecting KsgA and RrmA modifications and 18 h for detection of RsmE modification) with agitation at 200 rpm. Cells were then washed at 37 °C with agitation at 200 rpm in 30 µL wash buffer A for 15 min, 30 µL wash buffer B (0.9 M NaCl, 0.02 M Tris, pH 7.5, 0.01% SDS) for 15 min and then resuspended in 10 µL imaging buffer (0.025 M NaCl, 0.02 M Tris, pH 7.5). Then, 5 µL of this cell suspension was added to a coverslip (22 × 22 mm, thickness 0.13–0.17 mm, Menzel Gläser, pre-cleaned with argon plasma for at least 30 min (Femto Plasma Cleaner; Diener Electronic, Royal Oak, MI, USA) and subsequently attached to a Frame-Seal slide chamber (9 × 9 mm, Biorad, Hercules, CA, USA)), and a drop of mounting medium (Hydromount slide mounting medium, Fisher Scientific) was applied. Samples were then sealed with another plasma-cleaned coverslip and allowed to set for at least 30 min at room temperature before imaging.

**Fluorescence microscopy**. Two-color fluorescence images except those used to generate Supplementary Fig. 6 were acquired using a microscope previously described[42]. A HeNe laser (633 nm, 10 mW, 25-LHP-991-230, Melles Griot, Carlsbad, USA) and a diode laser operating at 488 nm (20 mW, PC13589, Cyan Scientific, Spectra Physics, Santa Clara, USA) were directed into an objective lens (Plan Fluor 20× air, numerical aperture (NA) 0.50, Nikon for all data except those in Supplementary Fig. 13, where two other objectives were used for comparison: CFI Plan Apochromat λ 40× air, NA 0.95 and 60× Plan Apo TIRF oil immersion, NA 1.45, both by Nikon) mounted on an Eclipse TE2000-U microscope (Nikon), parallel to the optical axis in epifluorescence mode. Emitted fluorescence was collected by the same objective and separated from excitation light by a dichroic mirror (FF500/646-DiO1; Semrock, Rochester, USA). Green and red fluorescence were separated from each other and filtered by a second dichroic mirror and filter sets (FF605-Di02, Roper Scientific, USA, FF03-525/50-25 (green emission), BLP01-635R-25 (red emission), all from Semrock), mounted on a Dual-View imaging system (Photometrics, Tucson, USA). Fluorescence emission was recorded on an EMCCD camera (Cascade II:512, Photometrics), cooled to −70 °C; each color was recorded on a separate half of the chip. Data (30–100 images per sample) were acquired in grids of four or five successive images (separated by 200 µm for images acquired using the 20× objective and 90 µm for images acquired using the 40× and 60× objectives), first under 633 nm illumination (with exposure times of 200 ms for KsgA data, 200 ms for RrmA data and 200 ms (20× objective), 50 ms (40× objective) or 25 ms (60× objective) for RsmE data), then under 488 nm illumination (with exposure times of 200 ms for KsgA data, 600 ms for RrmA data and 200 ms (20×), 50 ms (40×) or 25 ms (60×) for RsmE data) using a motorized stage (Optiscan system, Prior Scientific, Cambridge, UK), controlled with

Micromanager[43]. Out-of-focus images were discarded during acquisition, i.e., before analysis.

Three-color fluorescence images for multiplexed detection of methylations (Fig. 4 and Supplementary Fig. 11) and two-color images used to generate Supplementary Fig. 6 were acquired using another home-built microscope. The output from three lasers operating at 488 nm (Toptica, iBeam smart, 200 mW, Munich, Germany), 561 nm (Cobalt Jive, 200 mW, Cobalt, Sweden) and 640 nm (Coherent Cube, 100 mW, Coherent, USA) were attenuated using neutral density filters, and passed through quarter-wave plates, beam expanders and their respective excitation filters (LL01-488-25 for 488 nm FF01-561/14-25 for 561 nm and FF01-640/14-25 for 640 nm; Semrock). The lasers were combined using two dichroic mirrors (FF552-Di02-25×36, FF458-Di02-25×36, Semrock), and passed through the back port of a Nikon Ti-E Eclipse microscope (Nikon, Japan), where they were reflected down the optical axis of a 1.49 NA, 60× TIRF objective (UPLSAPO, 60XO TIRF, Olympus) in epifluorescence mode. Then, 641, 561 and 488 nm excitation was performed sequentially, and emitted fluorescence was collected by the same objective and separated from excitation light by a dichroic mirror (Di01-R405/488/561/635-25×36, Semrock). The red, yellow and green fluorescence emissions were filtered by further bandpass and/or longpass filters (BLP01-635R-25 (red emission), LP02-568RS-25 and FF01-587/35-25 (yellow emission), BLP01-488R-25 (green emission); Semrock) before being passed through a 2.5× beam expander and recorded on an EMCCD camera (Delta Evolve 512, Photometrics, AZ, USA) operating in frame transfer mode (EMGain of 11.5 e$^{-1}$ per ADU and 250 ADU per photon). Each pixel corresponded to a length of 131.5 nm. Data (100 images per sample) were acquired in grids of 10 by 10 successive images (separated by 100 µm), using a motorized stage (Proscan III system, Prior Scientific, Cambridge, UK), controlled with Micromanager[43]. Two-color images were first acquired under 640 nm illumination (with an exposure time of 10 ms), then under 488 nm illumination (with an exposure time of 40 ms), while three-color images were first acquired under 640 nm illumination (with an exposure time of 10 ms), then under 561 nm illumination (with an exposure time of 20 ms), and then under 488 nm illumination (with an exposure time of 10 ms). The microscope was fitted with a perfect focus system which auto-corrects the z-stage drift during a prolonged period of imaging. Out-of-focus images were discarded during acquisition, i.e., before analysis.

**Image analysis**. Bacteria detection (identifying regions of interest for analysis): Fluorescence images were analyzed using custom software written in MATLAB (MATLAB R2010b, The MathWorks, Inc., Supplementary Software). For each field of view, two or three images are acquired sequentially for each color channel, exciting the red ('red image') and then green ('green image') fluorophore, and, in the case of three-color imaging, the Cy3 fluorophore ('yellow image'). The process therefore begins with a step to ensure good registration across the image pairs or triplets. To do this, a sub-region is defined in the red image and the normalized two-dimensional cross-correlation coefficients are calculated between this sub-region and the yellow or the green image, from which the $xy$ position of maximal cross-correlation can be obtained for each image. The difference between the positions found gives the offset between the green or yellow channel from the red. In some instances, calculation of the cross-correlation does not return realistic values, in particular if the intensities are low or there are few image features. In this case, the offset is set to the median offset found for images of the given data set. The image of one channel is then shifted with respect to the other channel by this offset value, correcting for translational shift.

Prior to the detection of stained bacteria, which appear as local regions of high intensity above baseline levels in fixed positions within the field of view (Supplementary Fig. 4), the red image is converted to a binary image (mask) using a threshold intensity set to be the mean single pixel intensity of the bandpass-filtered red image plus 20 times this value. Pixels with intensity values below the threshold are set to zero and considered to be background, while pixels with values above the threshold are set to one and potential candidates for bacteria signals. This threshold was determined empirically and by visual inspection of the resulting mask and comparison with the original data. Only the signal from the red channel is used for bacteria detection to minimize any selection bias due to the expected large variation in fluorescence brightness in the green and yellow channels. For example, bacteria with highly methylated ribosomes might not give rise to sufficient intensity in the green channel to be detected, biasing detection toward bacteria with low ribosomal methylation state.

Selection of the signals from single, isolated bacteria: In the next step, objects (bacteria) are detected by tracing continuous regions of ones in the binary image (mask). It is necessary to include further selection criteria in the algorithm for robust discrimination against aggregates of bacteria which may consist of bacteria belonging to different strains. To this end, for each region, the position, size and continuity (number of pixels corresponding to "holes" with value zero in the region) is determined. Signals are assumed to correspond to single bacteria if the area of the region is 1–10 µm$^2$ and contains < 50 background pixels (value zero). Selection criteria based on region ellipticity were also explored but not found to improve discrimination. For regions that fulfill these requirements (single bacteria), the mean intensity for the region is calculated from the green, yellow and red original images and corrected for the local background by subtracting the average intensity of

a shell around the detected region (+/– 3px). The entire procedure is fully automated and does not require the user to select control areas or individual bacteria.

Analysis and presentation of the extracted data: Finally, the ratio of red/green intensity (or yellow/green in the case of three-color experiments) was calculated for each bacterium. Since the distribution of the two-color ratio for a single population is expected to be log-normal, the histograms were plotted logarithmically and fitted to a single-term Gaussian function, thus obtaining the center of each ratio distribution. All plots and further fits were created in OriginPro.

**Code availability**. MATLAB code for analyzing MR-FISH image data is available at https://github.com/kganzinger/Analysis-Software-for-in-situ-hybridization-data-in-single-cells. The code requires MATLAB R2010b (or later releases) and the following tool boxes to run: Image Processing, Curve Fitting and Statistics.

**Statistics**. Sufficient imaging data were collected to yield >450 cells per sample for analysis after thresholding, with three to seven replicate samples per experiment (with $n$ quoted in the appropriate figure legends). Descriptive statistics are used throughout.

**Data availability**. All imaging data used to produce the main text figures and supplementary figures, as well as Excel source data for main text Figs. 1–4 are deposited on figshare[44]. HPLC and thermal melting data are available on reasonable request.

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

## Acknowledgements

We thank Professor Peter L. Roach (University of Southampton) and Dr. Paul R. Race (University of Bristol) for their critical reading of the manuscript. This work was supported by the EU Innovative Medicines Initiative, IMI (RAPP-ID project, grant agreement, no. 115153), the UK Biotechnology and Biological Sciences Research Council, BBSRC (Project Grant: BB/J017906/1) and the UK Engineering and Physical Sciences Research Council, EPRSC (Project Grant: EP/M027546/1). M.H.H. was supported by a Junior Research Fellowship at Christ's College, University of Cambridge, and the Herchel Smith Foundation. J.W.C. and W.H.S. were supported by the Medical Research Council, UK (MC_U105181009 and MC_UP_A024_1008). D.K. is supported by the Royal Society.

## Author contributions

R.T.R. and M.R.C. conceived the MR-FISH approach. R.T.R., M.R.C., K.A.G., J.S. and D.K. designed experiments. B.W.L., C.S. and R.T.R. collected fluorescence melting data. R.T.R. and M.H.H. collected microscopy data. M.R.C. collected biochemical and microbiological data. R.T.R. and K.A.G. designed analysis software. K.A.G. wrote analysis

software. M.R.C., W.H.S., N.S. and J.W.C. contributed materials. R.T.R., M.R.C., K.A.G., B.W.L. and C.S. analyzed data. R.T.R., M.R.C., K.A.G., J.S. and D.K. wrote the paper.

## Additional information

**Competing interests:** The authors declare no competing financial interests.

