## [Peer Review File · Nature Communications]

Reviewers' comments:

Reviewer #1 (Remarks to the Author):

The manuscript by Ranasinghe et al describes a new and original method for single cell assessment (but not really precise measurements) of certain rare RNA modifications. Indeed, certain rare RNA modifications completely prevent Watson-Crick base-pairing and thus may be detected by altered hybridization properties. To my knowledge, this idea was already exploited for RNA-modification specific DNA chips, but not yet at the level of single cell microscopy. Single eukaryotic cell contains only 10-30 pg of total RNA (90% is rRNA) and this sensitivity cannot be achieved by conventional or even NGS-based techniques. Thus the proposed methodology represents an important contribution to the field of epitranscriptomics (new name for RNA modification) and provides a new insights to regulation and intrinsic heterogeneity of RNA modification in the cell.

The manuscript is clearly written and well illustrated, reference list is short, but covers sufficiently the field of research. Supplementary materials contain important optimization information of major experimental parameters. I am in favor of publishing this manuscript, with some modifications in discussion as indicated below.

Authors should clearly indicate limitations of the proposed method, which can be applied in its current form ONLY to a minor fraction of known RNA modifications, namely those preventing WC base-pairing. Only a few such residues are present in rRNA, which was a subject in this study.

Another limitation is probably the cellular content of a given RNA, rRNA represents 90% of total RNA and it would be good to know if modifications of other RNA species (tRNAs? others?) can be also detected by MR-FISH.

The third limitation is certainly global precision of methylation measurements. From the graph presented, the error seems to be rather high. What are the main reasons for such diversity? Since technical replicates demonstrate similar distribution to biological replicates, would this mean that something in sample preparation or hybridization conditions affects fluorescent measurements?

The title of the article is also somehow misleading: in the field, "epitranscriptome" term generally refers to mRNA modifications, while only rRNA was exploited in the current version of MR-FISH, I would clearly indicate rRNA methylation in the title, which corresponds well to real content.

Reviewer #2 (Remarks to the Author):

Ranasinghe et al. describes a new method, called MR-FISH, to detect RNA modifications in single cells. This method takes advantage of the disruption of base pairing caused by some RNA modifications. As a proof of concept, the authors examine prevalent and well-established RNA modifications in prokaryotic rRNA using their MR-FISH method. To demonstrate that MR-FISH can work for single cell analysis, they perform a mixing experiment. The authors also develop and provide scripts to automate the quantification of results from MR-FISH experiments. MR-FISH has the potential to be quite useful for the examination of RNA modification dynamics, and the authors provide convincing evidence of its ability to distinguish between modified and unmodified rRNA.

Currently, MR-FISH would have the greatest impact on the analysis of m6A modifications in single cells. Although the authors frame the advantages of their method within the context of the increasing

importance of m6A, they do not present experiments showing the efficacy of MR-FISH in detecting m6A modifications or for detecting other modifications on mRNA. The addition of these experiments would enhance the paper and help to justify the declaration of the utility for MR-FISH to detect epitranscriptomic modifications and not just rRNA modifications.

Major comments:

1) Testing of molecular beacon probes for an endogenous or in vitro transcribed mRNA with m6A modifications would greatly improve the utility of MR-FISH for the field. Ideally this would be done in eukaryotic cells. Another, less convincing, approach to demonstrate MR-FISH's application to m6A modifications would be to at least show an in vitro thermal melting experiment.

2) The advantage of MR-FISH over methods that pool the result of thousands of cells is its utility for single-cell analyses. However, only Figure 3 demonstrates the benefits of a single cell approach by showing the results of a mixing experiment. An additional demonstration of the utility of the single-cell approach would be helpful in differentiating the method from bulk analyses and demonstrating how the method could be used to address current challenges in the field.

Minor comments:

1) How does surrounding RNA structure, such as modifications that might be found in tRNAs, affect the ability of MB probes to hybridize to their target?

2) Labeling of rows in Figure 2 with the corresponding methylation being detected would improve readability.

3) Figure 3b, add 20%, 40%, etc. for easier readability.

4) Figure S3a define M and UM in figure legend.

5) Oligonucleotide sequences in Figure S1 should be presented as a table instead of a figure to allow easier access for future groups.

Reviewer #1 (Remarks to the Author):

The manuscript by Ranasinghe et al describes a new and original method for single cell assessment (but not really precise measurements) of certain rare RNA modifications. Indeed, certain rare RNA modifications completely prevent Watson-Crick base-pairing and thus may be detected by altered hybridization properties. To my knowledge, this idea was already exploited for RNA-modification specific DNA chips, but not yet at the level of single cell microscopy. Single eukaryotic cell contains only 10-30 pg of total RNA (90% is rRNA) and this sensitivity cannot be achieved by conventional or even NGS-based techniques. Thus the proposed methodology represents an important contribution to the field of epitranscriptomics (new name for RNA modification) and provides a new insights to regulation and intrinsic heterogeneity of RNA modification in the cell.

The manuscript is clearly written and well illustrated, reference list is short, but covers sufficiently the field of research. Supplementary materials contain important optimization information of major experimental parameters. I am in favor of publishing this manuscript, with some modifications in discussion as indicated below.

Authors should clearly indicate limitations of the proposed method, which can be applied in its current form ONLY to a minor fraction of known RNA modifications, namely those preventing WC base-pairing. Only a few such residues are present in rRNA, which was a subject in this study.

We have included a more detailed description of which bases are detectable by hybridisation in the discussion (see page 7, lines 17-22 in the revised manuscript), supported by one new reference (Ref. 29) and data (Fig. S12).

Another limitation is probably the cellular content of a given RNA, rRNA represents 90% of total RNA and it would be good to know if modifications of other RNA species (tRNAs? others?) can be also detected by MR-FISH.

We have now elaborated on this point in the discussion (page 7, line 30 - page 8, line 3 and supported by new references 30-33). We note that cellular content of a given RNA does not affect *whether* it can be detected by MR-FISH, but will determine the imaging mode required; mRNAs are often present at a handful of copies per cell, but many labs are able to detect these using single-molecule microscopy (e.g. refs 18-20, 32).

The third limitation is certainly global precision of methylation measurements. From the graph presented, the error seems to be rather high. What are the main reasons for such diversity? Since technical replicates demonstrate similar distribution to biological replicates, would this mean that something in sample preparation or hybridization conditions affects fluorescent measurements?

The variation of measured ratios will broadly depend on two sources of noise: i) stoichiometry of the fluorophores, which will be affected by hybridisation/sample preparation; and ii) signal noise. We have addressed this question explicitly with new experimental data (Figure S6 and its accompanying legend), which suggests that signal noise dominates the widths of two-color-ratio histograms.

The title of the article is also somehow misleading: in the field, "epitranscriptome" term generally refers to mRNA modifications, while only rRNA was exploited in the current version of MR-FISH, I would clearly indicate rRNA methylation in the title, which corresponds well to real content.

We agree that including the term “epitranscriptomic” could be misleading and have changed the title to “Detecting RNA base methylations in single cells by *in situ* hybridization”.

Reviewer #2 (Remarks to the Author):

Ranasinghe et al. describes a new method, called MR-FISH, to detect RNA modifications in single cells. This method takes advantage of the disruption of base pairing caused by some RNA modifications. As a proof of concept, the authors examine prevalent and well-established RNA modifications in prokaryotic rRNA using their MR-FISH method. To demonstrate that MR-FISH can work for single cell analysis, they perform a mixing experiment. The authors also develop and provide scripts to automate the quantification of results from MR-FISH experiments. MR-FISH has the potential to be quite useful for the examination of RNA modification dynamics, and the authors provide convincing evidence of its ability to distinguish between modified and unmodified rRNA.

Currently, MR-FISH would have the greatest impact on the analysis of m6A modifications in single cells. Although the authors frame the advantages of their method within the context of the increasing importance of m6A, they do not present experiments showing the efficacy of MR-FISH in detecting m6A modifications or for detecting other modifications on mRNA. The addition of these experiments would enhance the paper and help to justify the declaration of the utility for MR-FISH to detect epitranscriptomic modifications and not just rRNA modifications.

Major comments:

1) Testing of molecular beacon probes for an endogenous or *in vitro* transcribed mRNA with m6A modifications would greatly improve the utility of MR-FISH for the field. Ideally this would be done in eukaryotic cells. Another, less convincing, approach to demonstrate MR-FISH's application to m6A modifications would be to at least show an *in vitro* thermal melting experiment.

As discussed above, because of their low abundance, detecting endogenous mRNA (see response to reviewer #1 and clarified in the revised manuscript (page 7, line 30-page 8, line 3) requires a single-molecule microscopy variant of MR-FISH. While single-molecule FISH is now a reasonably widespread technique (e.g. refs 18-20, 32), implementing it in a timescale compatible with this manuscript revision was not feasible. We therefore addressed this question using thermal melting as suggested by the reviewer, which shows quantitatively that a single m⁶A modification destabilises duplex formation by a Molecular Beacon (Figure S12).

2) The advantage of MR-FISH over methods that pool the result of thousands of cells is its utility for single-cell analyses. However, only Figure 3 demonstrates the benefits of a single cell approach by showing the results of a mixing experiment. An additional demonstration of the utility of the single-cell approach would be helpful in differentiating the method from bulk analyses and demonstrating how the method could be used to address current challenges in the field.

We have now included a demonstration of an additional application of MR-FISH, to multiplexed detection of more than one methylation in a complex mixture (Page 6, lines 17 – 25, Figure 4 and S11). This greatly enhances the utility of MR-FISH for molecular diagnostics for antibiotic resistance, a key current challenge.

Minor comments:

1) How does surrounding RNA structure, such as modifications that might be found in tRNAs, affect the ability of MB probes to hybridize to their target?

This depends on the nature of the modification (as discussed in page 7, lines 15-20 in the revised manuscript); if there is a sequence where a strongly-destabilising modification is within 5-10 nucleotides of the base of interest, longer probes may be required.

2) Labeling of rows in Figure 2 with the corresponding methylation being detected would improve readability.

We have made this change.

3) Figure 3b, add 20%, 40%, etc. for easier readability.

We have made this change.

4) Figure S3a define M and UM in figure legend.

We have made this change.

5) Oligonucleotide sequences in Figure S1 should be presented as a table instead of a figure to allow easier access for future groups.

We have made this change (Table 1, page 12).

REVIEWERS' COMMENTS:

Reviewer #1 (Remarks to the Author):

The revised version of the manuscript by Ranasinghe et al generally include the majority of amendments proposed by reviewers. Title represents much better the content and important limitations of the method are now appropriately discussed. Some extra figures also bring additional important insights, like the influence of signal noise.

It is a bit disappointing that question on detection of other RNA species (like tRNAs) is only theoretically considered, without experimental validation with appropriate oligo.

Despite this I believe that manuscript is now ready for publication.

Reviewer #2 (Remarks to the Author):

The revised manuscript by Ranasinghe et al. is significantly improved and the responses addressed the concerns of the reviewers. I support publication in Nature Communications.